# Bayesian Network Analysis of Industrial Accident Risk for Fishers on Fishing Vessels Less Than 12 m in Length

Seung-Hyun Lee [1] , Su-Hyung Kim [1] , Kyung-Jin Ryu [2] and Yoo-Won Lee [2,*]

1   Training Ship, Pukyong National University, Busan 48513, Republic of Korea; leesh1225@pknu.ac.kr (S.-H.L.); prodoll1@naver.com (S.-H.K.)
2   Division of Marine Production System Management, Pukyong National University, Busan 48513, Republic of Korea; tuna@pknu.ac.kr
*   Correspondence: yoowons@pknu.ac.kr

**Abstract:** The Marine Stewardship Council estimates that approximately 38 million people worldwide work in fisheries, and more than one-third of the global population is dependent on aquatic products for protein, highlighting the importance of sustainable fisheries. The FISH Safety Foundation reports that 300 fishers die every day. To achieve sustainable fisheries as a primary industry, the safety of human resources is of the utmost importance. The International Maritime Organization (IMO) and the International Labor Organization (ILO) have made efforts towards this goal, including the issuance of agreements and guidelines to reduce industrial accidents among fishing vessel workers. The criterion for applying these guidelines is usually a total ship length $\geq$12 m or $\geq$24 m. However, a vast majority of registered fishing vessels are <12 m long, and the fishers of these vessels suffer substantially more industrial accidents. Thus, we conducted a quantitative analysis of 1093 industrial accidents affecting fishers on fishing vessels <12 m in length, analyzed risk using a Bayesian network analysis (a method proposed by the Formal Safety Assessment of the IMO), and administered a questionnaire survey to a panel of experts in order to ascertain the risk for different types of industrial accidents and propose specific measures to reduce this risk.

**Keywords:** Bayesian network analysis; fishing vessel; industrial accident; Formal Safety Assessment





## 1. Introduction

Fishing is recognized as one of the most hazardous occupations [1]. Maritime accidents frequently occur on fishing vessels due to various factors. Fishery workers face risks not only from the equipment, machinery, and structure of the vessel itself, but also from external dangers such as rough seas and severe weather conditions. Internationally, efforts have been made to improve safety standards for fishing vessels with a length of $\geq$24 m. These efforts include agreements such as the Torremolinos Convention and the Cape Town Agreement. Furthermore, in 2005, the IMO (International Maritime Organization), FAO (Food and Agriculture Organization), and ILO (International Labor Organization) proposed voluntary guidelines aimed at enhancing the safety of fishing vessels with lengths between 12 m and 24 m, a segment not covered by existing conventions.

However, in all countries where the United Nations Food and Agriculture Organization collects statistics, the number of registered powered or unpowered fishing vessels < 12 m exceeds the number of those $\geq$12 m [2]. South Korea, the focus of this study, is no exception. An analysis of accident compensation insurance data from the past five years (2018–2022), utilized in this study, reveals that although the proportion of crew members working on vessels less than 12 m is only 6.49%, the industrial accident rate stands at 8.01% (Figure 1). This indicates a disproportionately high industrial accident rate compared to the number of fishers.

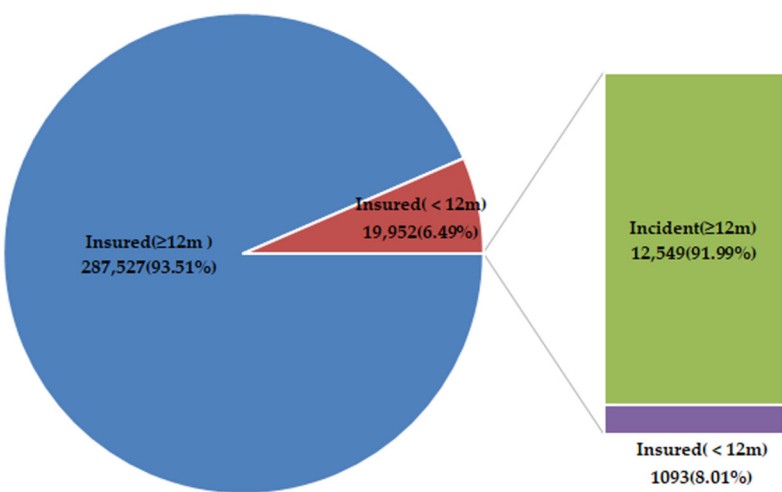

**Figure 1.** The recent 5-year (2018~2022) insured and incidents (≥12 m, and <12 m).

Extensive research has been conducted on the risk of fishing vessels, with recent studies actively utilizing Bayesian networks.

Jin et al. [3] performed logistic regression modeling on daily data for fishing vessel departures and accidents in coastal fishery regions in the Northeastern US over a period of 13 years, revealing a high likelihood of accidents during high wind speeds, with medium-sized vessels (51–150 t) showing the highest accident rates. Yu et al. [4] quantitatively evaluated the risk of ships operating in coastal waters based on ship type and operating areas from the perspective of maritime traffic using Bayesian network analysis. Cao et al. [5] aimed to identify the risk of maritime accidents such as collisions, sinkings, and hull damage based on factors such as total tonnage, engine output, and ship type using Bayesian network analysis and confirmed that fishing vessels and small ships have the highest risk. Wang et al. [6] quantitatively analyzed the risk of maritime accidents such as collisions, sinkings, etc., on fishing vessels based on factors including vessel length, total tonnage, ship type, engine output, and human factors using Bayesian network analysis. Obeng et al. [7] analyzed the quantitative risk of capsizing accident scenarios on small trawl fishing vessels with a length of less than 24 m by applying object-oriented Bayesian networks. Uğurlu et al. [8] utilized Bayesian networks to analyze accidents occurring on vessels longer than 7 m, confirming the significance of accident categories, vessel length, age, casualties, and vessel loss through chi-squared tests. Kim et al. [9] utilized Bayesian networks to analyze the risk of fatal accidents in trap fisheries, incorporating expert opinions and proposing improvement measures to reduce the risk of fatalities.

The Bayesian network utilized in previous studies is one of the risk analysis techniques proposed in the Formal Safety Assessment (FSA) of the International Maritime Organization (IMO) and has been widely used in the fields of reliability, risk analysis, and maintenance and repair. The ability to model probabilistic data with dependencies between events makes Bayesian belief networks suitable for risk analysis [10]. This characteristic also makes them suitable for modeling maritime accidents, enabling quantitative analysis of human and organizational factors [11]. To utilize Bayesian networks, prior probabilities need to be obtained. In this study, data on accident compensation insurance payouts for fishing vessels and fishers, implemented in South Korea since 2004, were utilized. This insurance has been in effect for approximately 20 years, and the accident payouts are deemed reliable as they are thoroughly investigated before being disbursed.

In this research, the length, which is an internationally recognized standard for vessel size rather than total tonnage or engine output, was utilized. Among these lengths, the focus was on vessels less than 12 m, which are not subject to international agreements and guidelines. Additionally, the study aimed to quantitatively analyze the risk of industrial accidents occurring during fishing operations, not only maritime accidents such as collisions, sinkings, and capsizing, but also industrial accidents that occur during fishing

operations. A Bayesian network proposed by the IMO's FSA was constructed, and actual industrial accidents affecting fishing vessel workers were analyzed using the chi-squared method. Finally, the opinions of an expert panel working on vessels less than 12 m were incorporated to analyze the major causes of industrial accidents on fishing vessels by type and propose improvement measures.

## 2. Materials and Methods

### 2.1. Materials

#### 2.1.1. Study Location (South Korea)

South Korea accounts for 2% of global marine capture production [2], and has active fisheries with over 65,000 registered fishing vessels each year (over 85% of all registered ships) [12]. Although South Korea is geographically a peninsula, its territorial waters share a border with North Korea. The main fishery sites are limited and use many of the same routes as commercial ships. Consequently, many industrial accidents occur at sea other than loss of life due to fishing activities. Thus, there is an urgent need to reduce the risk of accidents for small coastal fishing vessels.

#### 2.1.2. Study Vessels

In this study, small, coastal, commercial fishing vessels with a length of <12 m were selected. Wang et al. [13] confirmed the tendency of an increase in accident occurrence risk with the decrease in the length of fishing vessels, and Jin and Thunberg [1] reported that the risk of accidents is higher in coastal waters. Small fishing vessels of <12 m account for a high proportion of registered vessels, and frequently suffer industrial accidents. The focus was placed on these vessels due to the urgent need to reduce the high rate of accidents and to facilitate the efficient dissemination and utilization of the results in the future.

Meanwhile, in the process of adopting the Maritime Labor Convention 2006, which relates to commercial vessels, the ILO also introduced the Work in Fishing Convention 2007, which relates to fishing vessels, and defined the following relationships between the tonnage and lengths of fishing vessels. "Gross tonnages of 75, 300, and 950 gt are considered equivalent to lengths of 15, 24, and 45 m or overall lengths of 16.5, 26.5, and 50 m, respectively".

However, in Asia, including South Korea, which was the location for this study, the typical shape of fishing vessels differs from that in Europe, and ship sizes are defined in terms of tonnage. Most small commercial coastal fishing vessels in South Korea have a tonnage of <10 gt, meaning that the above criteria are not applicable. Among the various types of small, commercial, coastal fishing vessels in South Korea, the overall length of vessels with a tonnage of 2.99 gt was verified (Table 1) [14].

**Table 1.** Overall length characteristics of 2.99 gt fishing vessels.

| Type of Fishing Vessel | Overall Length (m) | | |
|---|---|---|---|
| | Average | Min | Max |
| Anchored stow net | 11.64 | - | - |
| Purse seine | 10.12 | 9.26 | 11.09 |
| Trap | 10.89 | 8.59 | 13.19 |
| Beam trawl | 10.75 | - | - |
| Gill net | 11.01 | 8.43 | 13.19 |
| Lift net | 10.71 | 10.38 | 11.36 |
| Multispecies fishing | 10.87 | 6.98 | 13.82 |
| Average | 10.86 | 8.73 | 12.53 |

As indicated by the data above, for the typical small, commercial, coastal fishing vessels in South Korea, the mean overall length at 2.99 gt is <12 m. Thus, vessels with a tonnage of <3 gt were considered to have an overall length of <12 m in this study. In addition, all terms relating to length used in this study refer to the overall length.

2.1.3. Accident Compensation Insurance Approval Data for Fishers

The data on the approval and payment of accident compensation for industrial accidents affecting fishers of small, commercial, coastal fishing vessels were obtained for the last 5 years, from 2018 to 2022. Since this is insurance compensation data for industrial accidents affecting fishers working on commercial fishing vessels, it includes relatively detailed information about the accidents, including the type, size, and age of the vessel, the identities of the fishers, and information about how the accident occurred. This makes it reliable data for the quantitative assessment of the risk of industrial accidents.

*2.2. Methods*

2.2.1. Quantitative Data Analysis

When investigating fishing vessel accidents, most accidents are found to occur during catch activities [15]. This is because, unlike commercial or other vessels, fishing vessels have to perform fishing activities while sailing. When comparing the safety logs of fishing vessels with other industrial groups, it is clear that this industry remains one of the most dangerous by a considerable margin [13].

Generally, personal accidents are not included in the statistics for accidents on fishing vessels, and tend to not be reported [16]. As a result, there is a shortage of data on the causes and outcomes of personal loss-of-life accidents aboard fishing vessels. This makes it difficult to quantitatively evaluate the risk of loss-of-life accidents. However, sufficient data on loss-of-life accidents on fishing vessels were acquired for this study. Based on these data, quantitative analysis was conducted depending on vessel length and age, human factors related to fishers, fishing processes, environmental factors, and accident types.

2.2.2. Risk Analysis Techniques

Bayesian networks, also known as belief networks, Bayes nets, or stochastic directed acyclic graphs, are a method for graphically expressing the joint probability distribution of a selected group of variables [17].

To apply a Bayesian network, it is first necessary to understand the logic of conditional probabilities. The conditional probability is the probability of a given event occurring when another event occurs. In other words, the probability of Event $A$ occurring when Event $B$ occurs is referred to as "the conditional probability of Event $A$ given Event $B$", which can be written as $P(A|B)$ (note that $P(A|B)$ can change depending on Event $B$, and generally $P(A|B)$ and $P(B|A)$ are not the same). The relationship between the prior and posterior probabilities of these two stochastic variables can then be used to define the relationship between the conditional and marginal probabilities (Equation (1)).

$$P(A|B) = \frac{P\,(A \cap B)}{P(B)} = \frac{P\,(B|A)P(A)}{P(B)} \tag{1}$$

where $P(A)$ is the prior probability of Event $A$ before Event $B$ occurring, $P(B)$ is the prior probability of Event $B$, $P(A \cap B)$ is the joint probability of both Event $A$ and Event $B$ occurring, $P(A|B)$ is the posterior probability of Event $A$ given Event $B$, and $P(B|A)$ is the likelihood function of Event $B$ given Event $A$ [18]. When this equation is used as a model to express the conditional probabilities between variables consisting of cause-and-effect nodes, the results can be visualized as a directed acyclic graph where the edges show dependencies between variables. This has the advantage of enabling predictions via backward chaining and prior probabilities. Recently, there has been growing interest in using this technique to model phenomena involving human and organizational factors [19]. Bayesian networks have been widely utilized as a modeling approach for constructing expert systems that

include uncertainty, and have been implemented in several studies related to marine traffic safety [19–21].

### 2.2.3. Chi-Squared Test

To verify the statistically significant relationships between each variable in the Bayesian network and the type of accident, which served as the final node, the chi-squared test was employed. The chi-squared test is based on whether there is a statistically significant difference between the observed and expected frequency. This test is used for qualitative data [22,23]. One of the main advantages of the chi-squared test of independence is that it can be applied not only to categorical data, but also to numerical data [24,25]. The general hypotheses of the chi-squared test of independence are shown below [23,24].

- Null hypothesis ($H_0$): The two variables are mutually independent.
- Alternative hypothesis ($H_1$ or $H_a$): The two variables are not mutually independent.

### 2.2.4. Expert Panel

There could be some risks that were not included in the data analysis results, and there could be differences between the risks included in the dataset and those experienced directly in the field. Hence, an expert panel consisting of 201 individuals from fishing vessels with an overall length <12 m was chosen to compare and integrate their opinions regarding the analyses described in Sections 2.2.1 and 2.2.2.

## 3. Results and Discussion

### *3.1. Analysis of Accident Compensation Insurance Data*

To infer the risk of industrial accidents for fishers, it was necessary to classify the data representing accident compensation payments in the last 5 years (2018–2022) to suit the subject and purpose of our analysis. In this section, accident compensation payment data for the target country and vessel types were utilized to quantitatively classify the occurrence of industrial accidents by vessel length and age, human factors, fishing processes, environmental factors, and accident types.

Note that the accident terminology adopted in this study, such as "stuck", "fall", "bump/hit", and "marine accident", are used officially as part of a code of industrial accidents [26], and have been maintained for ease of understanding and identification.

### 3.1.1. Vessel Length

Examined were the total number of accident compensation insurance subscribers for industrial accidents on fishing vessels, the total number of approved cases of accident compensation, the number of subscribers working on vessels of <12 m, and the number of approved cases of accident compensation for industrial accidents on vessels of <12 m (Table 2).

The number of insurance subscribers working on small, commercial fishing vessels of <12 m was 19,952 persons (approximately 6.49%) during the 5 years of our study. There were 13,642 cases (4.44%) of industrial accidents on all commercial fishing vessels, and there was a higher rate of industrial accidents on commercial fishing vessels of <12 m (1093 cases, 5.48%). There are several possible explanations for this, including the small working area on small vessels, loss of restoring force, greater effects of bad weather, and frequent leaving and entering of port due to fishing operations in nearby waters.

**Table 2.** Status of accident compensation insurance subscribers and industrial accidents (unit: persons).

|  | 2018 | 2019 | 2020 | 2021 | 2022 | Total |
|---|---|---|---|---|---|---|
| Number of insured fishers | 59,596 | 61,191 | 61,461 | 61,681 | 63,550 | 307,479 |
| Number of industrial accident incidents | 3063 | 2986 | 2578 | 2684 | 2331 | 13,642 |
|  | 5.14% | 4.88% | 4.20% | 4.35% | 3.67% | 4.44% |
| Number of fishers insured on vessels <12 m | 3752 | 3848 | 3944 | 4029 | 4379 | 19,952 |
|  | 6.30% | 6.29% | 6.42% | 6.53% | 6.90% | 6.49% |
| Number of industrial accident incidents on vessels <12 m | 186 | 229 | 231 | 230 | 217 | 1093 |
|  | 4.96% | 5.95% | 5.86% | 5.71% | 4.96% | 5.48% |

As mentioned previously, there is a trend of increased accident risk as vessel length decreases [13], and this can be verified in the table above. This is why focusing on vessels of <12 m was chosen. We analyzed the types and causes of industrial accidents aimed at mitigating this risk.

3.1.2. Vessel Age

Categorizing small, commercial, coastal fishing vessels of <12 m by vessel age, the analysis involved examining the number of industrial accidents relative to the number of insured vessels for each of the 5 years in this study (Table 3). It was observed that the most common vessel age was ≥20 years, with vessels of this age exhibiting the highest frequency of industrial accidents, totaling 379 cases (10.36%) over 5 years.

**Table 3.** Number of fishing vessels subscribed to accident compensation insurance and number of industrial accidents according to vessel age (units: vessels, events).

|  | Vessel Age | 2018 | 2019 | 2020 | 2021 | 2022 | Total |
|---|---|---|---|---|---|---|---|
| Number of insured vessels <12 m (Categorized by vessel age) | <5 | 469 | 478 | 451 | 395 | 382 | 2175 |
|  | ≥5 and <10 | 468 | 437 | 434 | 467 | 481 | 2287 |
|  | ≥10 and <15 | 433 | 457 | 468 | 514 | 553 | 2425 |
|  | ≥15 and <20 | 611 | 557 | 567 | 501 | 465 | 2701 |
|  | ≥20 | 517 | 635 | 715 | 820 | 970 | 3657 |
| Number of industrial accident incidents on vessels <12 m (Categorized by vessel age) | <5 | 30 | 37 | 28 | 27 | 13 | 135 |
|  |  | 6.40% | 7.74% | 6.21% | 6.84% | 3.40% | 6.21% |
|  | ≥5 and <10 | 26 | 34 | 31 | 56 | 39 | 186 |
|  |  | 5.56% | 7.78% | 7.14% | 11.99% | 8.11% | 8.13% |
|  | ≥10 and <15 | 37 | 31 | 38 | 45 | 41 | 192 |
|  |  | 8.55% | 6.78% | 8.12% | 8.76% | 7.41% | 7.92% |
|  | ≥15 and <20 | 47 | 44 | 52 | 26 | 32 | 201 |
|  |  | 7.69% | 7.90% | 9.17% | 5.19% | 6.88% | 7.44% |
|  | ≥20 | 46 | 83 | 82 | 76 | 92 | 379 |
|  |  | 8.90% | 13.07% | 11.47% | 9.27% | 9.49% | 10.36% |

Uğurlu et al. [8] previously reported that accident type showed significant relationships with both vessel length and vessel age. Our results also demonstrated a significant relationship between vessel age and industrial accidents. There are several possible explanations, but for old vessels of ≥20 years, we surmised that many loss-of-life accidents were caused by old, deteriorated fishing machinery and equipment. Other factors include decreased restoring force and corrosion of the hull due to aging.

### 3.1.3. Human Factors

Many studies have proposed that human error is a major contributing factor in marine accidents, and at least 80% of marine accidents are known to occur due to some form of human error [27–30]. When comparing the number of insured vessels with the number of insurance subscribers for small, commercial, coastal fishing vessels of <12 m (Table 4), it was found that the average fisher size was <2 persons per vessel. These vessels operate relatively close to the coast, suggesting a high risk of industrial accidents due to human error. Factors contributing to this risk include frequent port entry and exit, frequent travel between fisheries, and insufficient personnel during fishing activities. Such conditions can also impact response efforts in the event of an accident, potentially exacerbating the severity of urgent loss-of-life incidents.

**Table 4.** Numbers of fishing vessels <12 m and fishers enrolled in accident compensation insurance and average number of insured fishers per vessel (units: vessels, persons).

|  | 2018 | 2019 | 2020 | 2021 | 2022 | Total |
|---|---|---|---|---|---|---|
| Number of insured fishers on vessels <12 m | 3752 | 3848 | 3944 | 4029 | 4379 | 19,952 |
| Number of insured vessels <12 m | 2498 | 2564 | 2635 | 2700 | 2851 | 13,248 |
| Average number of fishers per vessel <12 m | 1.50 | 1.50 | 1.50 | 1.49 | 1.54 | 1.51 |

Wang et al. [13] noted that the presence of multinational fishers could impact maritime accidents, attributed to factors like language, education, and training disparities. The state of industrial accidents on small, commercial, coastal fishing vessels was analyzed based on both the nationality and gender of the fisher (Table 5).

**Table 5.** Status of accident compensation insurance subscribers and accident incidents by gender and nationality (unit: persons).

| Gender and Nationality | | 2018 | 2019 | 2020 | 2021 | 2022 | Total |
|---|---|---|---|---|---|---|---|
| Number of insured fishers | Male | 3252 | 3341 | 3429 | 3510 | 3844 | 17,376 |
| | Female | 500 | 507 | 515 | 519 | 535 | 2576 |
| | Nationals | 569 | 3641 | 3757 | 3859 | 4195 | 19,021 |
| | Foreigners | 183 | 207 | 187 | 170 | 184 | 931 |
| Number of industrial accident incidents | Male (including foreigners) | 165 | 209 | 201 | 203 | 195 | 973 |
| | | 5.07% | 6.266% | 5.86% | 5.78% | 5.07% | 5.60% |
| | Female (including foreigners) | 21 | 20 | 30 | 24 | 22 | 117 |
| | | 4.20% | 3.95% | 5.83% | 4.62% | 4.11% | 4.54% |
| | Nationals | 182 | 221 | 227 | 225 | 214 | 1069 |
| | | 5.10% | 6.07% | 6.04% | 5.83% | 5.10% | 5.62% |
| | Foreigners | 4 | 8 | 4 | 5 | 3 | 24 |
| | | 2.19% | 3.87% | 2.14% | 2.94% | 1.63% | 2.58% |

Unlike the findings of Wang et al. [13], our results showed that the rate of industrial accidents was much lower among foreign fishers compared to national (Korean) fishers. This is thought to be because, as we saw previously, for small, commercial, coastal fishing vessels of <12 m, the mean fisher size is <2 persons, and foreign insurance subscribers only accounted for around 4.66% (931 persons) of all subscribers over the 5 years of the study. Moreover, many accidents occur during the processes of casting and hauling fishing gear using fishing equipment, and these tasks are usually performed by highly experienced Korean fishers.

The classification based on gender revealed significant findings. In the case of females, both Korean and foreign, the majority were family members, and despite the significantly lower number of female subscribers overall, there were numerous industrial accidents affecting female fishers. The incidence of industrial accidents for females did not show a significant difference compared to males.

### 3.1.4. Fishing Processes

Unlike other commercial vessels, fishing vessels have to sail and perform fishing operations simultaneously. Small, commercial, coastal fishing vessels normally operate in relatively nearby fisheries. As such, there are often multiple work processes occurring simultaneously, as well as the frequent entering and leaving of ports and frequent loading and unloading, and there can be difficulties setting up equipment with small fisher sizes.

The fishing process was broadly categorized into sailing, fishing, maintenance, loading and unloading, and other. Out of the total 1093 cases of industrial accidents observed during the 5-year study period, over half (578 cases, 52.88%) occurred during fishing. Consequently, a significant number of accidents took place during fishing operations, with the analysis revealing that more industrial accidents occurred during maintenance compared to sailing (Figure 2). Note that the "other" category includes accidents where the precise time of the accident is unclear, such as disease while on board.

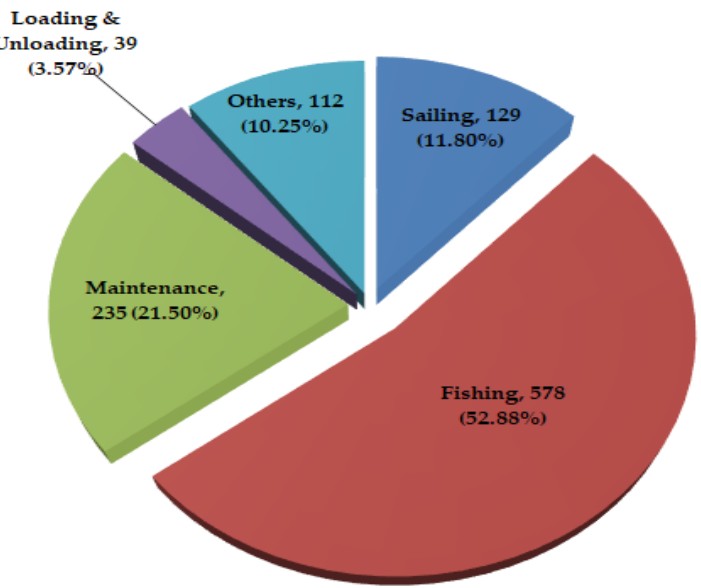

**Figure 2.** Status of industrial accidents among fishers categorized by fishing process (unit: persons, %).

### 3.1.5. Environmental Factors

During the final stage of an accident, for a given unsafe behavior to lead to an accident, the appropriate environmental factors must be present. Environmental factors include the weather, marine conditions, sailing type, time of day, traffic, fog, currents, and other factors external to the vessel structure. These environmental factors are beyond the scope of control of the vessel operators, and affect the movement of the vessel, which can be partially controlled by the vessel operators [8].

In this study, environmental factors were classified based on the details described in the records for accident compensation approval. The classifications included 'hull and fishing environment', 'fishing gear and fishing equipment', 'vessel machinery and equipment', 'weather and external marine environment', and 'other' (Figure 3).

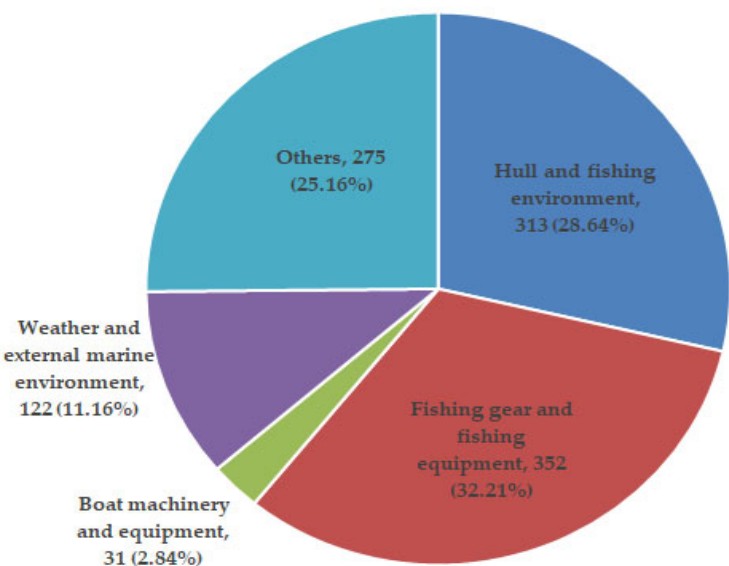

**Figure 3.** Status of industrial accident incidents based on environmental factors (unit: persons, %).

"Hull and fishing environment" include accidents due to a slippery floor, small spaces, or obstacles to movement in the hull or the fishing (working) environment. "Fishing gear and fishing equipment" include accidents due to all equipment used for fishing activities, including not only the fishing gear used directly for fishing itself, such as the fishing net and rope, but also the equipment used to move the gear to and from the sea, such as the caster and hauler. "Weather and external marine environment" include accidents due to external factors, such as poor weather, large waves, and strong currents, while "other" refers to accidents where the environmental factors were uncertain.

While there were some variations in the results, consistent trends for accidents due to similar environmental factors were generally observed. Particularly noteworthy was the finding that the most common environmental factor causing accidents across all 5 years of the study was 'fishing gear and fishing equipment' (352 cases, 32.21%).

Most loss-of-life accidents due to fishing gear and equipment involved part of the body, such as a hand or foot, getting caught in fishing equipment used for moving the gear to and from the sea, such as a caster or hauler. There were also accidents of various other forms, such as injury due to sudden snapping or breakage of gear under tension, or being knocked over by equipment moving on the deck.

The second-most common environmental factors in industrial accidents were the "hull and fishing environment". Most of the accidents due to the hull and fishing environment were slips or falls on a slippery deck, but there were also many accidents involving collisions with hull structures or falls due to narrow spaces or poor weather.

The "other" category included accidents caused by various environmental factors, but most common were impact accidents due to ship collisions. Other accident types included pain due to repeated movements or the use of excessive force, and injuries or disease due to enduring persistent or sporadic pain.

Because our study focused on small fishing vessels, there were a large number of accidents in the "weather and external marine environment" category due to the swaying of the ship in poor weather, such as strong winds and high waves. There were also many accidents caused by swaying in the wake of nearby large vessels.

3.1.6. Accident Types

In the insurance payment approval records, the type of accident is recorded in accordance with a code for the occurrence of industrial accidents [9]. The accident-type codes for small, commercial, coastal fishing vessels of <12 m in length include "trip/slip", "stuck (e.g., caster or hauler)", "bump/hit (e.g., fishing gear)", "fall (e.g., fishing port or overboard)",

"collapse (e.g., fishing gear)", "crushing", "unnatural posture (e.g., repeatedly in the same posture)", "exposure to abnormal temperatures (e.g., long-term exposure to sunlight)", "exposure to chemicals", "disease (e.g., working for long durations in a fishery)", and "other" [9]. Of these, vessel collision/sinking during vessel operations was classified as a "maritime accident", while industrial accidents occurring in each fishing process, other than maritime accidents and disease, were categorized as safety incidents, and the most frequent types of industrial accidents were further classified (Table 6).

**Table 6.** Status of industrial accidents among fishers categorized by accident type (unit: persons).

| Classification by Accident Type | | | 2018 | 2019 | 2020 | 2021 | 2022 | Total |
|---|---|---|---|---|---|---|---|---|
| Number of industrial accident incidents | Safety incident | Trip/Slip | 67 | 106 | 100 | 94 | 85 | 452 |
| | | | 36.02% | 46.29% | 43.29% | 40.87% | 39.17% | 41.35% |
| | | Bump/Hit | 26 | 25 | 32 | 35 | 20 | 138 |
| | | | 13.98% | 10.92% | 13.85% | 15.22% | 9.22% | 12.63% |
| | | Stuck | 22 | 18 | 19 | 19 | 16 | 94 |
| | | | 11.83% | 7.86% | 8.23% | 8.26% | 7.37% | 8.60% |
| | | Other | 27 | 33 | 40 | 28 | 31 | 159 |
| | | | 14.52% | 14.41% | 17.32% | 12.17% | 14.29% | 14.55% |
| | | Total | 142 | 182 | 191 | 176 | 152 | 843 |
| | | | 76.34% | 79.48% | 82.68% | 76.52% | 70.05% | 77.13% |
| | Marine accident | Collision/ Sinking | 18 | 27 | 26 | 28 | 23 | 122 |
| | | | 9.68% | 11.79% | 11.26% | 12.17% | 10.60% | 11.16% |
| | Disease | | 26 | 20 | 14 | 26 | 42 | 128 |
| | | | 13.98% | 8.73% | 6.06% | 11.30% | 19.36% | 11.71% |

An extensive array of industrial accidents was identified over the past five years, indicating substantial findings. Notably, during this period, 843 cases of industrial accidents were categorized as safety incidents occurring during fishing operations, comprising 77% of all industrial accidents. There were 122 and 128 cases (each 11%), respectively, of industrial accidents classified as maritime accidents (e.g., collision/sinking) and disease during vessel operations. Among safety incidents, "trip/slip" (452 cases, 41.35%), "bump/hit" (138 cases, 12.63%), and "stuck" (94 cases, 8.60%) showed higher frequencies than other accident types.

### 3.2. Bayesian Networks

Wang et al. [6], Obeng et al. [7], and Uğurlu et al. [8] confirmed variations in risk based on vessel age and human factors. Additionally, Kim et al. [9] observed differences in risk according to the process. This section aims to analyze the risk of industrial accidents based on nationality and gender as human factors, boat age, fishing processes, and environmental factors using Bayesian networks.

For the quantitative analysis, data classified as "other" and the disease data were excluded from the environmental factors and fishing processes due to their uncertain natures for maritime accidents. While the classification of the fishing processes was accurate, the environmental factor data were imprecise; therefore, these data were excluded from the environmental factor modeling. After excluding these factors, the inference of accident risk for the major safety incidents was conducted. Specifically, analysis was focused on the three most frequent types of safety incidents (trip/slip, bump/hit, and stuck). Prioritization for risk analysis was determined based on their highest incidence rates. Because the other types of accidents were extremely diverse, their incidences were too low for reliable quantitative inference.

To construct the Bayesian network, vessel age was classified as either $\geq 20$ years or <20 years, in accordance with Table 3. Nationality was classified as "national" or "foreigner", and gender as "male" or "female", as per Table 5. Probabilities for vessel age, nationality, and gender among all insurance subscribers were entered. Then, when connecting each parent node to the child node of industrial accident occurrence among small, commercial, coastal fishing vessels of <12 m, probabilities for vessel age ("$\geq 20$ years" or "<20 years"), nationality ("national" or "foreigner"), and gender ("male" or "female") among industrial accidents on small, commercial, coastal fishing vessels of <12 m were entered. While industrial accident occurrence on small, commercial, coastal fishing vessels of <12 m was the child node of each of the nodes described above, it was the parent node of environmental factors and fishing processes. In the process of linking the industrial accident node to its child nodes of environmental factors and fishing process, the prior probabilities were inputted, excluding the 'other' categories and uncertain data, as outlined above. Furthermore, in the process of connecting parent nodes to child nodes, the probabilities were calculated as conditional probabilities, as in Equation (1).

For the environmental factors, probabilities were assigned to 'hull and fishing environment', 'fishing gear and fishing equipment', 'vessel machinery and equipment', and 'weather and external marine environment' as causes of industrial accidents. Similarly, probabilities were assigned for the fishing processes.

At the terminal nodes representing the accident type, the Bayesian network models were completed by inputting the probabilities of each type of industrial accident as combinations of each environmental factor and fishing process (Figure 4), utilizing the Bayesian network software program 'Decision Science 2.3.5'.

To analyze the risk of industrial accidents on fishing vessels of <12 m, the Bayesian network model depicted in Figure 4 was constructed. The value of each node represents the quantitative percentage data analyzed in Section 3.1. Figure 5 demonstrates how the risk of each type of safety incident, represented by the leaf nodes, can be analyzed for native male fishers working on vessels aged $\geq 20$ years, using the data from Figure 4.

Utilizing the Bayesian network model in Figure 5, the risk of each type of accident was quantified based on vessel age, nationality, and gender (Table 7). It was found that the risk of industrial accidents was highest for native males working on vessels aged <20 years.

**Table 7.** Risk of accident types by vessel age, nationality, and gender.

| Vessel Age (Years) | Nationality | Gender | Trip/Slip | Bump/Hit | Stuck | Total |
|---|---|---|---|---|---|---|
| $\geq 20$ | Nationals | Male | 0.0042 | 0.0018 | 0.0019 | 0.0079 |
| $\geq 20$ | Nationals | Female | 0.00045 | 0.00019 | 0.00015 | 0.00079 |
| $\geq 20$ | Foreigners | Male | 0.000087 | 0.000038 | 0.00003 | 0.000155 |
| $\geq 20$ | Foreigners | Female | 0.000012 | 0.0000054 | 0.0000042 | 0.0000216 |
| <20 | Nationals | Male | 0.0077 | 0.0033 | 0.0026 | 0.0136 |
| <20 | Nationals | Female | 0.00097 | 0.00042 | 0.00033 | 0.00172 |
| <20 | Foreigners | Male | 0.00019 | 0.000083 | 0.000065 | 0.000338 |
| <20 | Foreigners | Female | 0.000012 | 0.0000054 | 0.0000042 | 0.0000216 |

The risk of each type of accident was also quantified based on fishing process and environmental factors, utilizing the Bayesian network model in Figure 5 (Table 8). Accidents due to fishing gear during fishing were most common, and "bump/hit" and "stuck" accidents were more common than other accident types. During sailing, maintenance, and loading/unloading, "trip/slip" accidents due to the "hull" showed the highest risk.

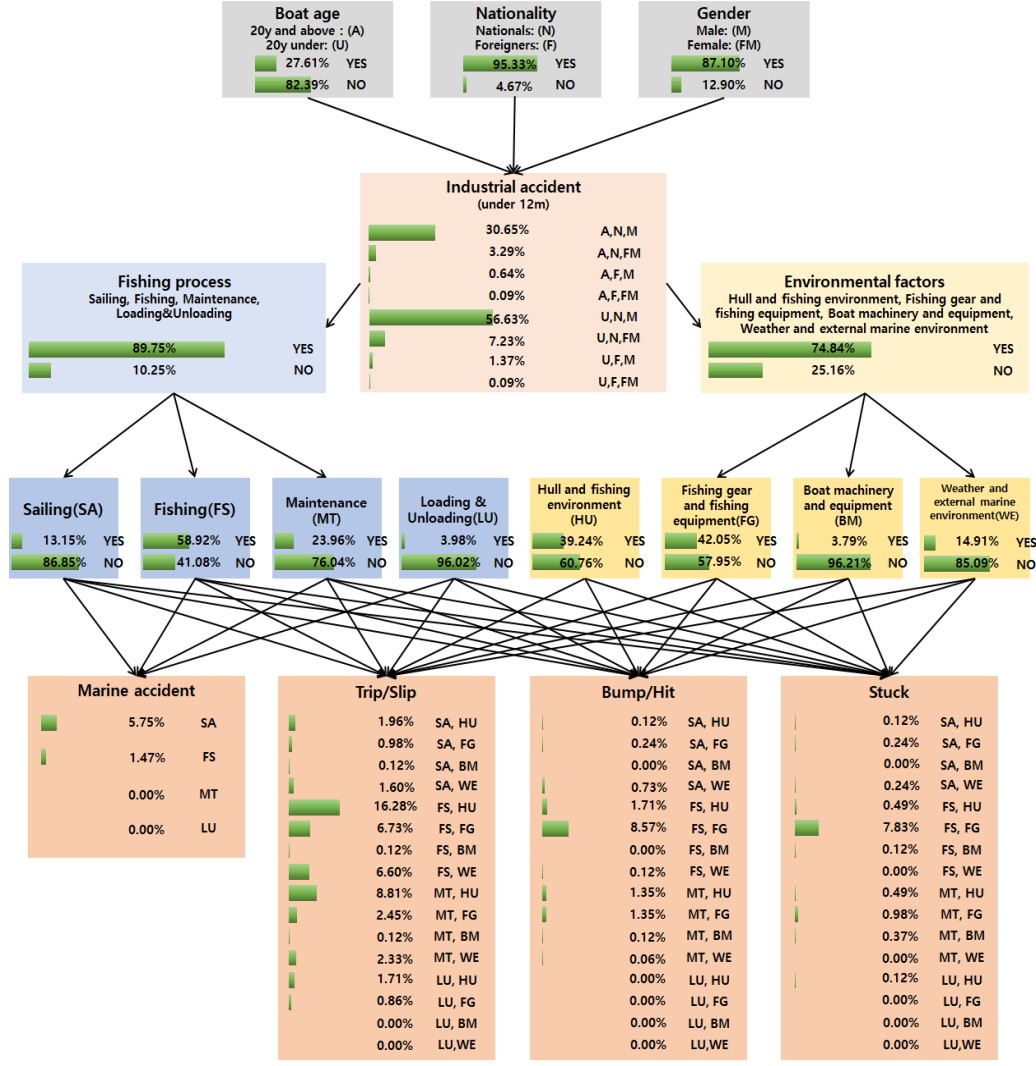

**Figure 4.** Bayesian network model for analysis of industrial accidents among fishers.

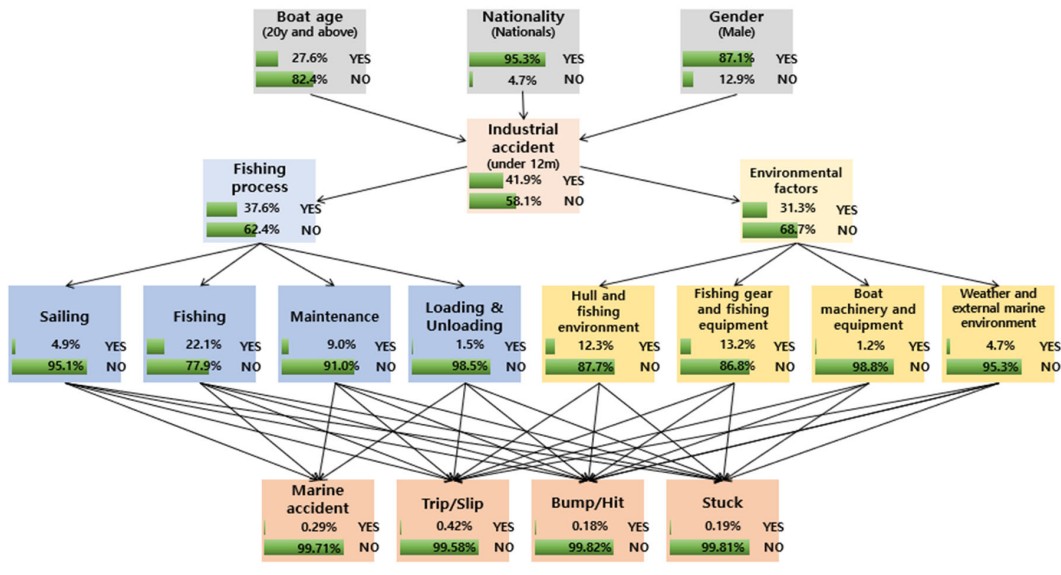

**Figure 5.** Bayesian network model for analyzing the risk of industrial accident among native male fishers employed on fishing vessels aged ≥20 years.

**Table 8.** Risk of accident types based on fishing processes and environmental factors.

| Fishing Process | Environmental Factor | Trip/Slip | Bump/Hit | Stuck | Total |
|---|---|---|---|---|---|
| Sailing | Hull and fishing environment | 0.02 | 0.0012 | 0.0012 | 0.0224 |
| | Fishing gear and fishing equipment | 0.0098 | 0.0024 | 0.0024 | 0.0146 |
| | Vessel machinery and equipment | 0.0012 | 0 | 0 | 0.0012 |
| | Weather and external marine environment | 0.016 | 0.0073 | 0.0024 | 0.0257 |
| Fishing | Hull and fishing environment | 0.163 | 0.017 | 0.0049 | 0.1849 |
| | Fishing gear and fishing equipment | 0.067 | 0.086 | 0.078 | 0.231 |
| | Vessel machinery and equipment | 0.0012 | 0 | 0.0012 | 0.0024 |
| | Weather and external marine environment | 0.066 | 0.012 | 0 | 0.078 |
| Maintenance | Hull and fishing environment | 0.088 | 0.013 | 0.0049 | 0.1059 |
| | Fishing gear and fishing equipment | 0.025 | 0.013 | 0.0098 | 0.0478 |
| | Vessel machinery and equipment | 0.0012 | 0.0012 | 0.0037 | 0.0061 |
| | Weather and external marine environment | 0.023 | 0.0061 | 0 | 0.0291 |
| Loading and unloading | Hull and fishing environment | 0.017 | 0 | 0.0012 | 0.0182 |
| | Fishing gear and fishing equipment | 0.0086 | 0 | 0 | 0.0086 |
| | Vessel machinery and equipment | 0 | 0 | 0 | 0 |
| | Weather and external marine environment | 0 | 0 | 0 | 0 |

*3.3. Chi-Squared Test*

Ten hypotheses were formulated to discern the relationships between the type of industrial accident, as analyzed using the Bayesian network in Section 3.2, and the vessel age, fisher gender and nationality, fishing processes, and environmental factors (Table 9). Chi-squared tests were conducted to verify each hypothesis, utilizing SPSS 21.0 for data analysis from 817 cases out of the total 1093 industrial accidents, as outlined in Section 3.2.

**Table 9.** Chi-squared test hypotheses and significance values for accident type, age, gender, nationality, fishing processes, and environmental factors.

| | Hypothesis | Significance | Result |
|---|---|---|---|
| $H_0$ | There is no significant relationship between accident type and vessel age | 0.733 | Accepted |
| $H_1$ | There is a significant relationship between accident type and vessel age | | Rejected |
| $H_2$ | There is no significant relationship between accident type and gender | 0.041 * | Rejected |
| $H_3$ | There is a significant relationship between accident type and gender | | Accepted |
| $H_4$ | There is no significant relationship between accident type and nationality | 0.149 | Accepted |
| $H_5$ | There is a significant relationship between accident type and nationality | | Rejected |
| $H_6$ | There is no significant relationship between accident type and fishing process | 0.000 * | Rejected |
| $H_7$ | There is a significant relationship between accident type and fishing process | | Accepted |
| $H_8$ | There is no significant relationship between accident type and environmental factors | 0.000* | Rejected |
| $H_9$ | There is a significant relationship between accident type and environmental factors | | Accepted |

\* $p < 0.05$ represents a significant relationship.

3.3.1. Chi-Squared Test Results

According to the results of the chi-squared tests, there was no significant relationship between vessel age and industrial accident type ($p = 0.733$, $p > 0.05$), meaning that $H_0$

was accepted and $H_1$ was rejected. In other words, vessel age and accident type are not directly related.

There was a significant relationship between gender and industrial accident type ($p = 0.041$, $p < 0.05$), meaning that $H_3$ was accepted and $H_2$ was rejected. In other words, accident type differed depending on the gender of the fisher. Although the result of the chi-squared test was significant, given that the number of female fishers was much lower than the number of male fishers, this result could be considered to have low reliability.

There was no significant relationship between nationality and industrial accident type ($p = 0.149$, $p > 0.05$), meaning that $H_4$ was accepted and $H_5$ was rejected. In other words, fisher nationality and accident type are not directly related.

There was a significant relationship between fishing process and industrial accident type ($p = 0.000$, $p < 0.05$), meaning that $H_7$ was accepted and $H_6$ was rejected. In other words, fishing process and accident type were directly related.

There was a significant relationship between cause of accident and industrial accident type ($p = 0.000$, $p < 0.05$), meaning that $H_9$ was accepted and $H_8$ was rejected. In other words, accident cause and accident type were directly related. Notably, we observed relationships between trips, weather, and hull environment, with weather and hull environment affecting trip accidents.

Chi-squared tests were conducted to analyze the relationships between the three main accident types ("trip/slip", "bump/hit", and "stuck") with fishing process and environmental factors (Table 10).

**Table 10.** Chi-squared test hypotheses and significance values for the three main accident types (trip/slip, bump/hit, and stuck) and the fishing processes and environmental factors.

| | Hypothesis | Significance | Result |
|---|---|---|---|
| $H_{10}$ | There is no significant relationship between the three main accident types (trip/slip, bump/hit, and stuck) and fishing processes | | Accepted |
| $H_{11}$ | There is a significant relationship between the three main accident types (trip/slip, bump/hit, and stuck) and fishing processes | 0.733 | Rejected |
| $H_{12}$ | There is no significant relationship between the three main accident types (trip/slip, bump/hit, and stuck) and environmental factors | | Rejected |
| $H_{13}$ | There is a significant relationship between the three main accident types (trip/slip, bump/hit, and stuck) and environmental factors | 0.041 * | Accepted |

* $p < 0.05$ represents a significant relationship.

According to the results of the chi-squared tests, there was a significant relationship between fishing processes and industrial accident type (trip/slip, bump/hit, and stuck; $p = 0.000$, $p < 0.05$), meaning that $H_{11}$ was accepted and $H_{10}$ was rejected. In other words, fishing process and accident type (trip/slip, bump/hit, and stuck) were directly related.

There was also a significant relationship between accident cause and industrial accident type (trip/slip, bump/hit, and stuck; $p = 0.000$, $p < 0.05$), meaning that $H_{13}$ was accepted and $H_{14}$ was rejected. In essence, there was a direct relationship between accident cause and accident type (slip/trip, bump/hit, and stuck). Particularly noteworthy was the observed association between trips and hull environment, indicating that hull environment influenced trip accidents.

3.3.2. Correspondence Analysis for the Chi-Squared Test Results

The results of the chi-squared tests showed significant relationships between accident type (trip/slip, bump/hit, and stuck), fishing processes, and accident cause. Consequently, correspondence analysis was conducted to elucidate the specific relationships between fishing processes and accident causes. In Figures 5 and 6 below, a shorter distance between the row and column points in each dimension can be interpreted as a stronger relationship.

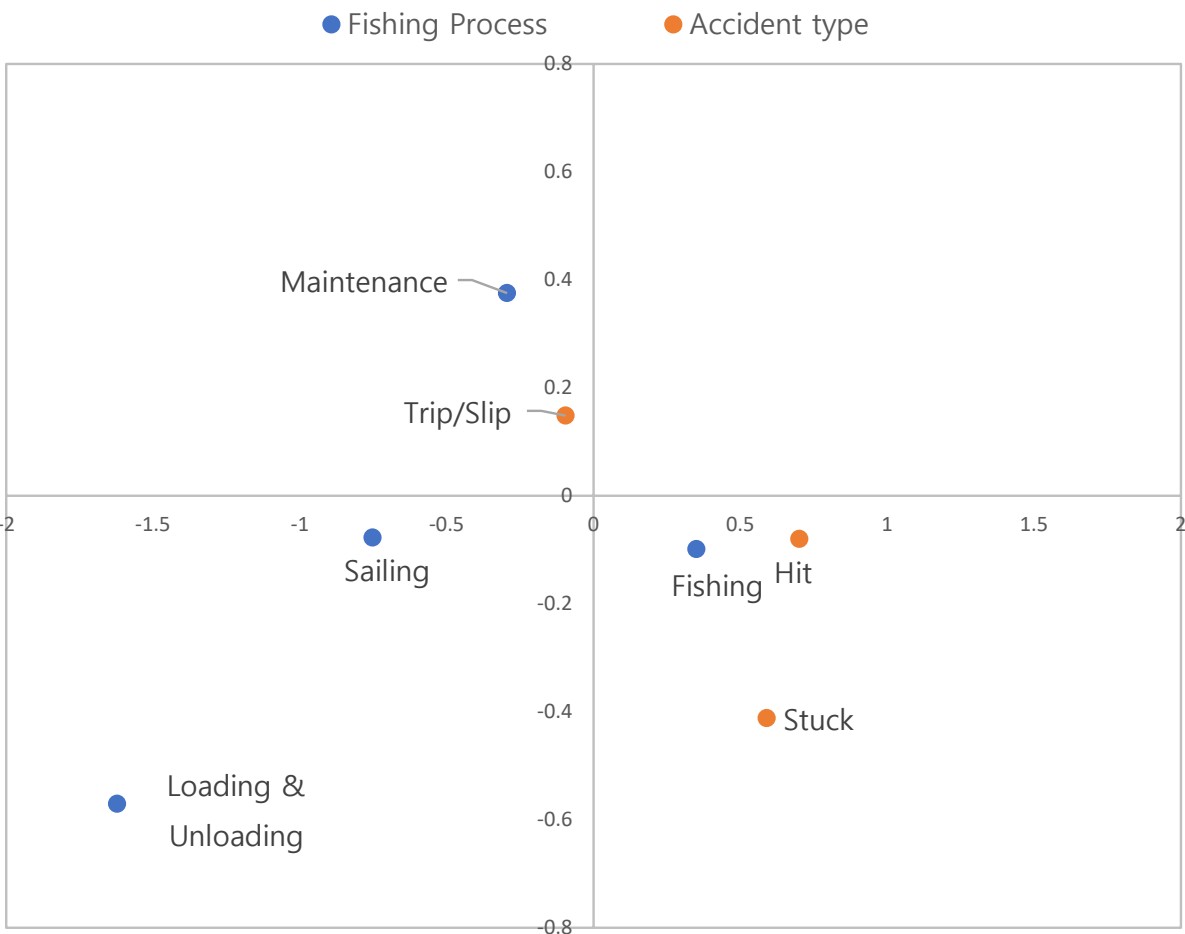

**Figure 6.** Chi-squared test correspondence analysis for fishing processes and accident type.

Figure 6 shows the results of correspondence analysis of the fishing process and accident type in each dimension. The accident types showing the strongest relationship, per fishing process, were "trip/slip" for "sailing" and "maintenance", and "hit" for "fishing". Specifically, the trip/slip accident type was related to the sailing and maintenance processes. Among the processes, the fishing stage showed a strong relationship with the hit accident type, but also showed stronger relationships with trip/slip and stuck accidents than with other processes. Thus, Figure 6 shows that, among the processes, fishing shows stronger relationships with the various accident types (trip/slip, bump/hit, and stuck) than the other processes, suggesting that this process exposes fishers to the risk of diverse accidents. In addition, trip/slip accidents show relationships with various processes, suggesting that they have the highest risk among the accident types.

Figure 7 shows the correspondence analysis for the relationships between environmental factors and accident type. Here, "bump/hit" and "stuck" accident types are associated with fishing gear and fishing equipment, while "trip/slip" accidents are associated with the hull and fishing environment. The integration of Figures 6 and 7 reveals that improvements in fishing processes and environmental factors could be introduced to mitigate the incidence of the main three accident types.

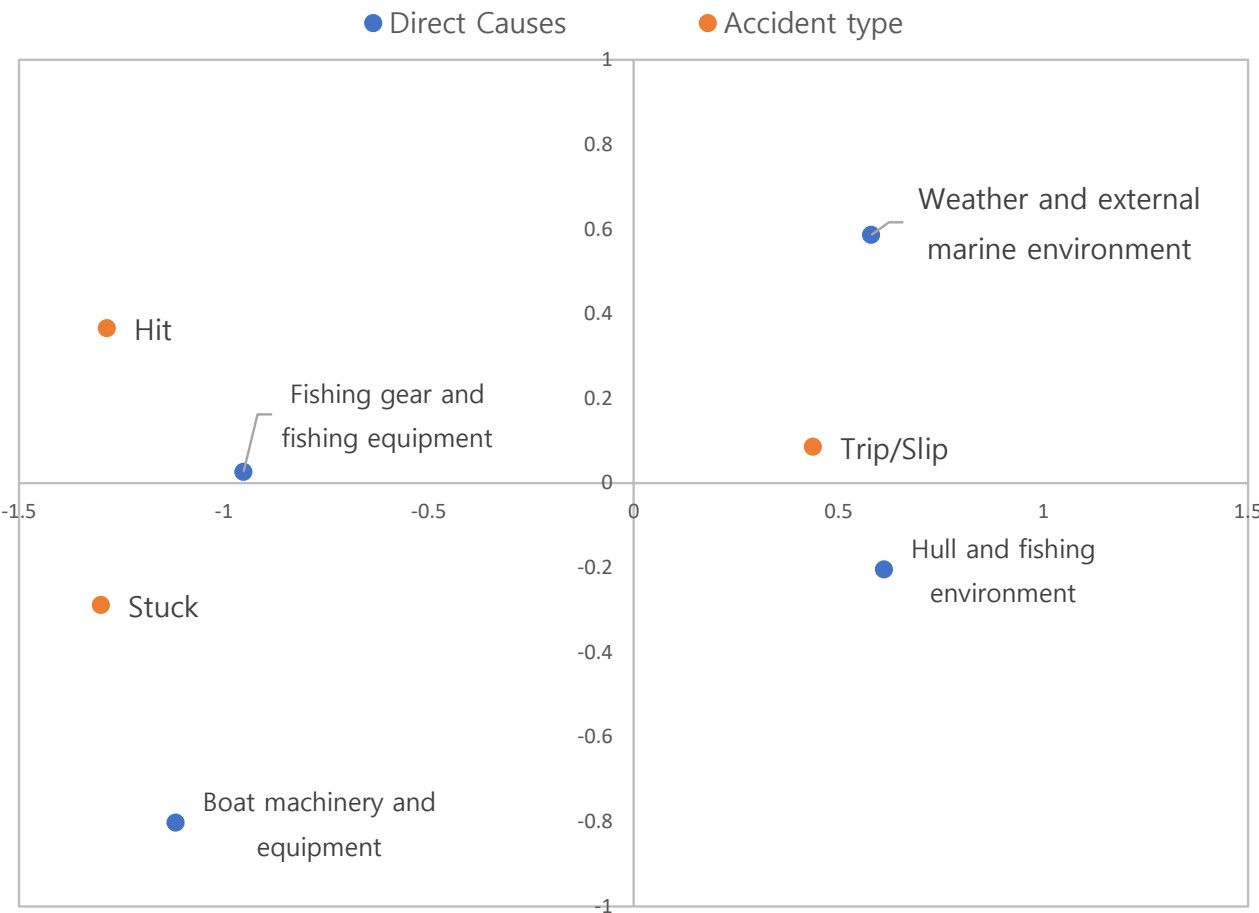

**Figure 7.** Chi-squared test correspondence analysis for direct causes and accident types.

### 3.4. Expert Panel Opinions

The reason that Bayesian networks are suitable for marine modeling is because of their ability to utilize expert knowledge when there is a shortage of data. Bayesian networks are a way of expressing domain knowledge, and explicitly include probabilistic dependence and causal relationships between major factors [31]. Bayesian network learning is considered to be evidence-based because the models are constructed from actual accident data [32]. The objective was to compare the analysis of actual industrial accident data with the insights provided by a panel of experts.

Opinions were gathered from an expert panel through a questionnaire survey administered to fishers on vessels of <12 m. A total of 201 responses were received, comprising 189 males and 12 females. The average fishing experience was 18 years, with 47 participants having encountered an actual industrial accident. The expert panel was deemed adequately reliable for this study.

#### 3.4.1. Fishing Processes

Regarding the fishing processes quantified in Section 3.1.4 and analyzed using Bayesian networks in Section 3.2, the experts were tasked with categorizing the risk of industrial accidents for each process as either "high risk", "moderate risk", or "low risk" (Table 11).

**Table 11.** Aggregation of risk perception responses from expert panel categorized by fishing processes (unit: persons).

| Classification by Fishing Process | | Sailing | Fishing | Maintenance | Loading and Unloading |
|---|---|---|---|---|---|
| Risk classification | High risk | 6 | 69 | 2 | 3 |
| | | 2.99% | 34.33% | 1.00% | 1.49% |
| | Moderate risk | 61 | 87 | 22 | 24 |
| | | 30.35% | 43.28% | 10.95% | 11.94% |
| | Low risk | 127 | 38 | 170 | 167 |
| | | 63.18% | 18.91% | 84.58% | 83.08% |
| | No response | 7 | 7 | 7 | 7 |
| | | 3.48% | 3.48% | 3.48% | 3.48% |

As per the experts' assessment, the fishing process was identified as having the highest risk of industrial accidents, aligning with the outcome observed in our analysis. In other words, to reduce the overall risk of industrial accidents, it would be most efficient to eliminate or reduce the risk of accidents during fishing. In addition to fishing, the expert panel's evaluation of the risk during other processes also showed similar trends to our analysis.

3.4.2. Environmental Factors

The expert panel's opinions regarding the risk of industrial accidents stemming from the various environmental factors quantified in Section 3.1.5 and analyzed using Bayesian networks in Section 3.2 were collected (Table 12).

**Table 12.** Aggregation of risk perception responses from expert panel categorized by environmental factors (unit: persons).

| | Classification by Environmental Factors | Response | |
|---|---|---|---|
| Expert panel survey | Harsh natural conditions | 63 | 31.34% |
| | Communication issues due to noise during fishing | 40 | 19.900% |
| | Cramped workspace | 36 | 17.91% |
| | Fishing gear under tension | 27 | 13.43% |
| | Cluttered workspace | 11 | 5.47% |
| | Other | 24 | 11.94% |
| | Total | 201 | 100.00% |

According to the expert panel, the highest risk was associated with harsh natural conditions, such as high waves, followed by communication issues due to noise during fishing, a cramped workspace, fishing gear under tension, and a cluttered workspace. These findings showed some differences from our analysis. This discrepancy is likely attributed to the fact that the data only recorded the single cause with the greatest effect on industrial accident occurrence, whereas actual accidents often result from a combination of factors.

Additionally, we examined the expert panel's opinions concerning the risks associated with fishing gear and equipment, which constituted the highest proportion of industrial accidents among the environmental factors quantitatively analyzed in Section 3.1.5 (Table 13).

**Table 13.** Aggregation of risk perception responses from expert panel categorized by fishing gear-related factors (unit: persons).

| | Classification Based on Fishing Gear-Related Factors | Response | |
|---|---|---|---|
| | Aging fishing equipment | 54 | 26.87% |
| | Absence of warning devices | 39 | 19.40% |
| | Absence of emergency shutdown devices | 37 | 18.41% |
| Expert panel survey | Structural defects in fishing equipment | 35 | 17.41% |
| | Absence of protective devices in fishing equipment | 18 | 8.96% |
| | Other | 18 | 8.96% |
| | Total | 201 | 100.00% |

According to the expert panel, the most common causes of industrial accidents due to fishing gear and equipment were, in descending order, aging fishing equipment, the absence of warning devices, the absence of emergency shutdown devices, structural defects in fishing equipment, and the absence of protective devices in fishing equipment. Therefore, leveraging the insights of the expert panel, we pinpointed the specific risk factors associated with fishing gear and equipment, the environmental factor accountable for the highest number of industrial accidents as per the accident compensation approval data.

3.4.3. Causes and Problems

The expert panel was also consulted regarding causes and issues associated with industrial accidents that were not covered in the approval process for accident compensation insurance payouts, the primary data source for this study (Table 14).

**Table 14.** Aggregation of risk perception responses from expert panel categorized by causes and issues related to industrial accidents (unit: persons).

| | Classification by Causes and Issues | Response | |
|---|---|---|---|
| | Neglect of safety in fishing practices | 79 | 39.30% |
| | Insufficient pre-training on risk factors and situations | 46 | 22.89% |
| Expert panel survey | Absence of work regulations and guidelines | 28 | 13.93% |
| | Inadequate health management of fishers | 18 | 8.96% |
| | Other | 30 | 14.93% |
| | Total | 201 | 100.00% |

The accident causes and problems highlighted by the expert panel were, in descending order, neglect of safety in fishing practices, insufficient pre-training on risk factors and situations, the absence of work regulations and guidelines, and inadequate health management of fishers.

More than half of the experts identified human error as a cause of industrial accidents, including the fishing convention of focusing on the catch over safety, and the lack of pre-training on risk factors and situations. The four most commonly mentioned causes and problems were items that could be improved at a personal or institutional level.

**4. Conclusions**

In this study, the quantitative analysis of the risk of industrial accidents occurring on vessels under 12 m in length was pursued. Utilizing trustworthy data from accident compensation insurance for fishing vessels and fishers, an industrial accident database

was constructed, aiming to identify risk factors. Furthermore, chi-squared tests were conducted to enhance the reliability of the Bayesian network constructed. Finally, through the perspectives of an expert panel working in the field, the causes and issues of industrial accidents were analyzed to increase the practicality of the research.

Using Bayesian networks, the risk associated with age, nationality, and gender was found to be the highest among male domestic fishers under the age of 20 working on boats for less than 20 years. Conversely, the lowest risk was observed among female foreign fishers aged 20 and over working on boats for less than 20 years. The analysis revealed that the risk of industrial accidents occurring was approximately 629 times higher for female foreign fishers under the age of 20 working on boats for less than 20 years compared to their counterparts. However, chi-squared tests indicated that age and nationality of fishers were not significantly associated with the occurrence of industrial accidents on boats.

On boats under 12 m, sailing and fishing occur within cramped workspaces, requiring maintenance, loading, and unloading to be performed with a limited number of fishers. Analyzing the risks posed by environmental factors classified into hull and fishing environment, fishing gear and fishing equipment, boat machinery and equipment, and weather and external marine environment, industrial accidents attributed to fishing gear and fishing equipment during fishing were found to be the highest. The second-highest risk was also attributed to hull and fishing environment during fishing, confirming the quantified values of heightened risk during this process. Experts also analyzed that fishing carried a high level of risk.

The risk associated with industrial accidents varied depending on their nature, with trip/slip being the highest during fishing due to the hull and fishing environment, showing approximately an eight times higher risk compared to sailing and maintenance for the same factors. Expert opinions compiled suggest that factors such as cramped workspace and cluttered workspace could be contributing to these incidents.

In the fishing process, bump/hit and stuck were most commonly associated with fishing gear and fishing equipment. This resulted in bump/hit being approximately five times higher and stuck approximately sixteen times higher in risk compared to hazards arising from the hull and fishing environment within the same process. These incidents often occur due to equipment such as rollers used to haul fishing gear on most small vessels and the strong tension applied to the gear. Expert opinions suggest that factors such as aging fishing equipment, absence of warning devices, and absence of emergency shutdown devices could be contributing to these incidents.

While this study conducted an objective analysis based on data, there are still several limitations, such as the focus on vessels under 12 m in length in Korea. This may limit its applicability to analyzing industrial accidents among fishers in other regions due to regional and cultural differences. Additionally, while the environmental factors of industrial accidents are clearly specified, it is challenging to account for multiple complex causes when only one cause is specified. However, by incorporating quantitative risks for each operational process and environmental factor, along with expert opinions aimed at reducing industrial accidents, and by minimizing unnecessary expenditures and enhancing regulatory efficiency, a more effective system can be established [33].

**Author Contributions:** Conceptualization, S.-H.L.; methodology, K.-J.R.; software, S.-H.L.; analysis, S.-H.K.; writing—original draft preparation, Y.-W.L.; writing—reviewing and editing, K.-J.R.; supervision, Y.-W.L. All authors have read and agreed to the published version of the manuscript.

**Funding:** This research was conducted as part of the "Development and demonstration of data platform for AI-based safe fishing vessel design (20220210)" of the Ministry of Oceans and Fisheries.

**Institutional Review Board Statement:** Not applicable.

**Informed Consent Statement:** Not applicable.

**Data Availability Statement:** The data used to support the findings of this study are available from the corresponding author.

**Conflicts of Interest:** The authors declare no conflicts of interest.

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
