# Peer review of "Bayesian Network Analysis of Industrial Accident Risk for Fishers on Fishing Vessels Less Than 12 m in Length"

_sustainability, doi:10.3390/su16103977_

Round 1
Reviewer 1 Report
Comments and Suggestions for Authors
his paper conducted a quantitative analysis of 1,093 industrial accidents affecting fishers on fishing vessels and analyzed risk using a Bayesian network analysis (a method proposed by the Formal Safety Assessment of the lMO). Then it proposes specific measures to reduce this risk. The full paper is somewhat innovative, but there are still some questions that the author needs to answer.
1. The introductory part does not clearly introduce the mechanism of the method used in this paper and lacks comparison with other methods. The advantages of the method used in this paper should be highlighted.
2. Please explain how Eq.(1) corresponds to the below methods.
3. Please explain how are these aspects selected, such as vessel length, vessel age, human factors, fishing process, environmental factors?
4. Some tables can be displayed in the form of bar charts to make the effect more obvious.
5. The fonts in the picture are not clear enough, such as Fig.1.
6. The grammar of the full paper requires correction by the author. References should cite recent works to highlight the significance of this work.
Comments on the Quality of English Languagehis paper conducted a quantitative analysis of 1,093 industrial accidents affecting fishers on fishing vessels and analyzed risk using a Bayesian network analysis (a method proposed by the Formal Safety Assessment of the lMO). Then it proposes specific measures to reduce this risk. The full paper is somewhat innovative, but there are still some questions that the author needs to answer.
1. The introductory part does not clearly introduce the mechanism of the method used in this paper and lacks comparison with other methods. The advantages of the method used in this paper should be highlighted.
2. Please explain how Eq.(1) corresponds to the below methods.
3. Please explain how are these aspects selected, such as vessel length, vessel age, human factors, fishing process, environmental factors?
4. Some tables can be displayed in the form of bar charts to make the effect more obvious.
5. The fonts in the picture are not clear enough, such as Fig.1.
6. The grammar of the full paper requires correction by the author. References should cite recent works to highlight the significance of this work.
Author Response
첨부파일을 참조해 주시기 바랍니다.

Reviewer 2 Report
Comments and Suggestions for Authors
The manuscript contains interesting findings. The Authors used a Bayesian network for quantitative analysis and risk analysis of industrial accidents on vessels less than 12 m in length involving fishermen. In addition, the chi-square test was used to analyze actual industrial accidents involving fishermen on fishing vessels. Additionally, the Authors, based on the opinions of a panel of experts, analyzed the main causes of various types of industrial accidents on fishing vessels and proposed preventive actions aimed at reducing the number of industrial accidents on fishing vessels <12 m in length. The scope of the work coincides with the subject of the special issue to whose job was reported. There are publications in the literature on the analysis of the causes of fishing vessel accidents (also using Bayesian networks and the chi-square test) or the contribution of the human factor to fishing vessel accidents.
The introduction is extensive, but the work lacks a clearly formulated purpose of the work. The described methodology is consistent with applicable standards. The Authors presented an analysis of accident insurance data based on factors such as vessel length, vessel age, human factors, fishing processes, environmental factors and accident types. Moreover, based on human, physical and environmental factors and fishing processes, a Bayesian network of industrial accidents on fishing vessels <12 m in length was constructed. The Authors quantified the risk of each type of accident depending on the age of the vessel, nationality and gender as well as depending on the fishing process. and environmental factors. Additionally, the Authors defined research hypotheses and conducted a chi-square test to verify them, and also compared the results of the analysis of actual data on industrial accidents with the opinions of a panel of experts.
The manuscript is written well and clearly, the experimental part is quite well prepared, as are the results, which are well described and related to the research conducted. The selected literature is relevant, although as many as 72% of the bibliography items are older than the last 5 years.
Reviewer 3 Report
Comments and Suggestions for Authors
This article analyzes the risks of<12 meter fishing boats. It has important research value and significance.
I think improvements can be made in the following areas:
(1) Can the data for 2021 be updated in the second paragraph of the introduction? On the other hand, I think it would be better to explain the proportion of<12 meter fishing boat accidents to the total number of fishing boat accidents.
(2) In the introduction, the explanation of why this issue is being studied is not thorough enough. What is the progress of risk research on<12 meter fishing boats by other scholars? What is the difference between this article and existing research?
(3) A graphical summary or table can be added to the introduction to better illustrate why this issue is being studied.
(4) A common risk method used by Bayesian analysts has led to a lack of innovation in this article. Existing scholars have made many improvements to Bayesian analysis, and this article can apply the improved Bayesian analysis. Or propose a new improved Bayesian method.
(5) The conclusion is written relatively simply and not deep enough. It is recommended to expand.
(6) The proposed countermeasures are detached from the conclusion, and the written countermeasures are common and not targeted enough. Strengthening investment in emergency management is also a very important strategy. Suggest citing this paper on security investment: DOI10.1016/j.jlp.2023.105230.
(7) There has been relatively little literature in the past three years, so it is recommended to update it.
Round 2
Reviewer 3 Report
Comments and Suggestions for Authors
accept